# Uncommon Variants of Mature T-Cell Lymphomas (MTCLs): Imaging and Histopathologic and Clinical Features with Updates from the Fourth Edition of the World Health Organization (WHO) Classification of Lymphoid Neoplasms

**DOI:** 10.3390/cancers13205217

**Published:** 2021-10-18

**Authors:** Ahmed Ebada Salem, Yehia H. Zaki, Gamal El-Hussieny, Khaled I. ElNoueam, Akram M. Shaaban, Bhasker Rao Koppula, Ming Yang, Mohamed Salama, Khaled M. Elsayes, Matthew F. Covington

**Affiliations:** 1Department of Radiology and Imaging Sciences, Utah University School of Medicine, Salt Lake City, UT 84132, USA; Ahmed.Salem@utah.edu (A.E.S.); Akram.Shaaban@hsc.utah.edu (A.M.S.); Bhasker.Koppula@hsc.utah.edu (B.R.K.); Matthew.Covington@hci.utah.edu (M.F.C.); 2Department of Radio Diagnosis, Faculty of Medicine, Alexandria University, Alexandria 21526, Egypt; Yehia.Zaki@alexmed.edu.eg (Y.H.Z.); Khaled.Elnoueam@alexmed.edu.eg (K.I.E.); 3Department of Medical Oncology and Nuclear Medicine, Faculty of Medicine, Alexandria University, Alexandria 21526, Egypt; Jamal.attia@alexmed.edu.eg; 4Division of Radiology, Mayo Clinic, Phoenix, AZ 85054, USA; Yang.ming@mayo.edu; 5Department of Pathology, Mayo Clinic, Rochester, MN 55905, USA; Salama.Mohamed@mayo.edu; 6Department of Diagnostic Radiology, The University of Texas MD Anderson Cancer Center, Houston, TX 77030, USA

**Keywords:** updates, 2016 WHO classification of hematologic malignancies, mature-T-cell lymphoma, imaging, FDG-PET/CT

## Abstract

**Simple Summary:**

Familiarity with the updated fourth edition WHO classification of lymphoid malignancies released in 2016, and the new terminology introduced, is crucial for oncologists, pathologists and radiologists. It is mandatory to be aware of rare variants of T-cell lymphomas, specifically mature T-cell lymphomas, including clinicopathologic keys and the most common imaging features and sites of involvement for each subtype. Even though some of these disorders may have perceptible clinical and imaging features, they may overlap with more common disorders, causing delay in diagnosis and treatment. Understanding the appropriate clinical setting and imaging findings will help radiologists to include these disorders in their differential diagnosis. Imaging plays a pivotal role in subclassifying these subtypes of MTCLs, affecting prognosis and treatment implications. Many of these disorders if diagnosed early can be potentially treatable, and early, aggressive intervention may be lifesaving.

**Abstract:**

Understanding the pathogenesis and molecular biology of malignant lymphomas is challenging, given the complex nature and incongruity of these disorders. The classification of lymphoma is continually evolving to account for advances in clinical, pathological, molecular biology and cytogenetic aspects, which impact our understanding of these disorders. The latest fourth edition of the WHO classification of lymphoid malignancies was released in 2016 to account for these changes. Additionally, unlike B-cell lymphomas (BCL), T-cell lymphomas (TCL) are uncommon, and may be sporadically experienced in clinical practice. These disorders are rare, thus early diagnosis is challenging for both physicians and radiologists, owing to the overlap in clinical and imaging features with other, more common disorders. We aim to discuss some rare variants of T-cell lymphomas, including clinicopathologic and imaging features, as well as to give a glimpse of the updates contained within the new 2016 WHO classification.

## 1. Introduction

Lymphomas can be generally classified into Hodgkin’s lymphoma (HL) and non-Hodgkin’s lymphomas (NHL), and either B-cell or T-cell lymphoma, based on their cellular origin. Virtually all systems can be involved including nodal and extranodal sites. However, in HL, at onset, nodal and splenic involvement is more common, while extranodal manifestation is more commonly seen in NHL [1]. The incidence of extranodal involvement in NHL has been continuously rising due to many factors including immunosuppression and increased use of chemotherapy [2,3]. Historical advances in our understanding of lymphoma behavior and pathogenesis have been restricted largely to BCLs and HL compared with mature TCLs (MTCLs). The cellular origin of MTCL is complex and not fully understood. T cells emerge from the bone marrow and experience maturation within the thymus by undergoing T-cell-receptor (TCR) gene rearrangement. Diversity of MTCLs is attributed to different types of T cells, each secreting different sets of cytokines signaling other cascades of immune cells. T cells include T-helper (Th) cells, suppressor T cells, cytotoxic cells, memory cells, γδ T cells and natural killer (NK) cells, which is considered a subtype of T cell [4]. The most recent WHO 2016 classification of lymphoid tumors now recognizes 29 discrete types of MTCLs. According to the initial site of involvement, MTCLs can clinically be broadly classified into one of four categories, including (1) leukemic disease; (2) nodal disease; (3) extranodal disease; and (4) cutaneous disease. The most common MTCLs subtypes include PTCL, not otherwise specified (NOS) and angioimmunoblastic T-cell lymphoma (AITL), other subtypes include hepatosplenic gamma delta T-cell lymphoma (HSTCL), anaplastic large-cell lymphoma (ALCL), and subcutaneous panniculitis-like T-cell lymphoma (SPTCL), the primary T-cell lymphoma of the intestine [5]. Generally, cutaneous T-cell lymphoma (CTCL) without distant metastatic disease is associated with a favorable prognosis of a 90% 5-year survival rate. Later in the course of the disease, progression, as manifested by nodal and visceral involvement, can be seen with a markedly decreased 5-year survival to almost 0%. Prognosis of most PTCL subtypes is generally poor [6].

Imaging in general, and specifically FDG PET/CT, plays an essential role in the management of lymphomas. The use of FDG-PET/CT can change the tumor staging in almost 30% of cases. FDG-PET/CT can impact management by further subclassifying lymphomas clinically into aggressive or indolent subtypes based on standardized uptake values (SUVs) and heterogeneity of disease [7]. Aggressive lymphomas often display higher SUV (>10–13) on the baseline, or focal disease in nodal or extranodal sites, or diffuse disease within extranodal sites such as the spleen, liver, or bone marrow [8]. FDG PET/CT can also help with detecting transformation of lymphomas into more aggressive counterparts by detecting an increase in SUVs of lymphomatous masses despite adequate treatment received, directing the physician to change the management and treatment plan. FDG-PET has been shown to be useful in identifying active residual sites of disease by demonstration of persistent metabolic activity with those masses [9,10]. In general, both TCLs and BCLs are FDG-avid on PET/CT. Although FDG avidity has been found to correlate with aggressiveness and clinical behavior of BCLs, the relationship between FDG uptake and different TCL subtypes has not been unquestionably settled. TCLs usually have a poorer prognosis compared to BCLs with a 5-year survival rate of less than 20% [6].

## 2. Changes Contained in the Updated 2016 WHO Classification of T-Cell Lymphoid Malignancies

The updated 2016 World Health Organization (WHO) classification of tumors of hematopoietic and lymphoid tissue (Table 1) [11] account for significant changes in our understanding of lymphoma behavior including both B-cell and T-cell lymphomas since 2008, and refines the pathogenesis and molecular biology of different types of lymphoma and lymphoma-like disorders [5,12]. Significant advances have occurred in the classification of TCLs, which has prompted revisions in the 2016 updated WHO classification (Table 2). TCLs can be broadly divided into peripheral (PTCL) and cutaneous types (CTCL). Compared to BCLs, TCLs are relatively rare neoplasms ranging from indolent to aggressive lymphomas. TCLs account for less than 1% and nearly 12% of all lymphomas and NHL cases, respectively [13]. Compared to BCL, TCLs have a more aggressive clinical course and are more challenging to diagnose and treat given their rarity.

CTCL are considered indolent lymphomas, and comprise diverse disorders defined by uncontrolled proliferation of monoclonal proliferation of T cells within the skin, skin appendages and some mucosal sites. They generally represent 6% of all NHL, and major subtypes include mycosis fungoides (MF)/Sezary syndrome (SS), primary cutaneous anaplastic large-cell lymphoma (pcALCL) and lymphomatoid papulosis (LyP). PTCL, on the contrary, are considered aggressive lymphomas. The more modern WHO classification has updated the mature T- and NK-cell neoplasms classification [5] (Table 3). Both clinicians and radiologists should also be aware of the pathogenesis of these disorders, pathologic findings, most common clinical and imaging features and most reasonable differential diagnosis for each disorder presented (Table 4). The purpose of this article is to describe the updates and new nomenclature contained within the new fourth edition of the WHO classification of lymphoid neoplasms, including the new terminology used and the reclassification of former entities. These changes are implemented to reflect the advancement of our understanding of lymphoma molecular biology.

## 3. Uncommon Mature T-Cell Lymphoma Variants

### 3.1. Subcutaneous Panniculitis-like T-Cell Lymphoma (SPTCL)

Subcutaneous panniculitis-like T-cell lymphoma (SPTCL) is a rare form of MTCL originating from mature cytotoxic cells. The first case was reported by Gonzalez et al., in 1991 [14]. Distinction from the more aggressive primary cutaneous gamma delta TCL was made in the updated WHO classification [5].

SPTCL accounts for less than 1% of NHL and has a slight female prediction. It occurs in both children and adults, with an average age of onset of 36 years [15]. Lesions present as red or violaceous nodules, plaques, or ulcerated lesions. It was previously believed to be associated with a poor prognosis; however, the 5-year survival rate is now known to approach 80%. Pathologically, subcutaneous panniculitis-like T-cell lymphoma is seen as lymphocytic infiltration surrounding subcutaneous fat lobules; this is usually associated with karyorrhexis, or fat necrosis. SPTCL is characterized by T-cell infiltration of the subcutaneous fat without dermal or epidermal involvement, as in MF and SS. Autoimmune diseases are seen with approximately 20% of cases, and lupus erythematosus panniculitis (LEP) is usually part of the differential diagnosis due to similar clinical and histologic features [16].

Clinical courses have been reported: a prolonged course of recurrent panniculitis or rapid clinical deterioration secondary to hemophagocytic syndrome (HPS) [17]. On imaging, SPTCL appears as enhanced hypermetabolic lesions localizing to the subcutaneous tissues. The differential diagnosis can be broad according to imaging and clinical findings and includes many inflammatory panniculitides associated with rheumatologic disorders (i.e., SLE), infectious nodules, and cutaneous metastasis from melanoma or breast cancer [18]. The nodules can progress and ulcerate. Clinically, SLE nodules present with preferential involvement of the proximal extremities and face, contrarily, SPTCL involves the lower and upper extremities as well as the trunk (Figure 1). Almost 50% of patients will manifest with systemic manifestations including fever, weight loss, pancytopenia, and elevated liver enzymes. Symptoms are often due to nodal involvement and bone marrow with hemophagocytic syndrome (HPS), and it is usually associated with a fatal outcome. The role of imaging, mainly utilizing FDG PET/CT, is to exclude visceral involvement and delineate the extent of disease for improved clinical management [19].

### 3.2. Anaplastic Large-Cell Lymphoma (ALCL)

In the new 2016 WHO classification, ALCL now has different cytogenetic subsets that may have prognostic implications on overall survival [5]. ALCL originates from T-helper 17. Systemic ALCL can be divided based on the presence of anaplastic lymphoma kinase gene (ALK) enzyme into positive and negative variants [20]. ALK-positivity is of great significance for clinical management and prognosis. ALK-positive cases have a better prognosis compared to ALK-negative cases, with a more than 70% 5-year survival rate being common for ALK-positive cases [19,21].

Historically, systemic ALCL was diagnosed as an undifferentiated carcinoma or malignant histiocytosis before the discovery of specific tumor antigen Ki-1 and specific chromosomal translocations that result in the expression of ALK protein seen in this rare subtype of lymphoma [18].

Clinically, three different types of ALCL can be differentiated based on initial site of presentation: systemic ALCL, primary cutaneous ALCL, and breast implant-associated ALCL (BIA-ALCL). Cutaneous ALCL is considered an indolent tumor, presenting in older age groups (Figure 2).

Systemic ALCL presents with disseminated disease including generalized lymphadenopathy and involvement of multiple extranodal sites including bone, lung, mediastinum, pleural pericardium and the gastrointestinal tract (Figure 2 and Figure 3). It may also demonstrate peritoneal lymphomatosis, and/or omental and mesenteric masses, resembling disseminated DLBCL [19].

Imaging may be requested to differentiate between cutaneous and systemic ALCL AKL variants. FDG PET/CT has a particularly important role in differentiating between the systemic ALCL AKL-positive variant that demonstrates higher uptake compared with ALK-negative ALCL cases [19]. Differentiation is important as it has consequential prognosis and treatment implications [22] (Figure 3).

### 3.3. Breast Implant-Associated Anaplastic Large-Cell Lymphoma (BIA-ALCL)

BIA-ALCL is a rare type of NHL and a subtype of ALCL. It is a new provisional entity distinguished from other ALK ALCLs introduced in the most recent WHO classification of lymphoid malignancies. It is considered a noninvasive disease associated with excellent outcome [5]. All cases of BIA-ALCL are of the ALK-negative variety, expressing CD30. The first case of BIA-ALCL was described in 1997 [23]. Despite being first described in 1997, diagnosis of BIA-ALCL remains uncommon and delayed, often only made at the time of implant revision surgery performed for a persistent seroma, and thus radiologists need to be aware of this entity. In 2011, an FDA warning was released about a possible risk of developing BIA-ALCL for women with breast implants regardless of the specific implant type. It is estimated that the lifetime risk of BIA-ALCL in women with breast implants is 1 in 30,000 [24,25].

The two described histopathological aspects of these tumors include in situ BIA-ALCL (confined to implant capsule), mostly presenting with delayed implant effusion. The second form is infiltrative BIA-ALCL (infiltrating the capsule) mostly related to mass-forming lesions years after surgery [26]. Both clinicopathologic forms may coexist in the same patient. In case of late peri-implant seroma, aspiration of at least 50 mL must be performed and sent for culture, cytology and flow cytometry. Fine needle aspiration cytology samples play an important role, and cell block is also crucial as they provide the basis for immunohistochemistry to evaluate for BIA-ALCL versus other entities such as infections/scar/granuloma formation. In cases of mass-forming lesions, core needle should be the preferential approach, and cores should be submitted in saline for flow cytometry if BIA-ALCL is a diagnostic consideration [24]. Morphology of cells shows large, atypical lymphoid cells with inflammatory background and eccentric, kidney-shaped nuclei, with homogeneous eosinophilic cytoplasm “hallmark cells” [27]. Cells express CD30 and frequent markers for cytotoxic T-cells. The precise pathogenic mechanisms of this type of lymphoma are still to be established. Chronic bacterial antigen stimulation or chronic irritation from implant characteristics may play a role in tumorigenesis [28].

BIA-ALCL patients commonly present with unilateral breast enlargement, due to late peri-implant effusion, or less commonly peri-implant mass, and/or axillary mass with worse prognosis if mass lesions are present. The differential diagnosis for peri-implant effusions includes a delayed peri-implant seroma, hematoma, infection, and/or implant rupture. A small volume of peri-implant fluid may also be physiologic. Onset of new breast enlargement months to years after breast implant placement with imaging demonstrating a peri-implant effusion should prompt evaluation for BIA-ALCL. Peri-implant seromas are more common in the immediate postoperative period, and persistence or development of seromas after one year is considered rare. If a mass lesion is present, whether in the breast or axillary regions, biopsy of the mass lesion should also be performed with core needle biopsy [29].

On MRI, seromas display T2 fluid hyperintense signals. MRI with silicone selective sequences, when appropriate, may be helpful to exclude an implant capsular rupture as a cause of the peri-implant fluid. Peri-implant hematomas may also be seen in the immediate postoperative period. Following surgery, hematomas may also be seen following trauma or in the setting of a known coagulopathy. In cases of biopsy-proven or high suspicion of BIA-ALCL, patients should undergo PET/CT to evaluate for presence of hypermetabolic capsular masses, chest wall invasion, especially if there is high clinical suspicion for disease spreading outside of the breast prior to any surgical intervention, because postoperative intervention will lead to false-positive results. Moreover, suspicious locoregional adenopathy can be detected preoperatively by imaging and sampling during surgery [24]. FDG PET/CT findings in BIA-ALCL may show a hypermetabolic peri-implant effusion and enhanced peri-implant masses with or without axillary nodes (Figure 4 and Figure 5). Localized BIA-ALCL confined to the breast implant capsule often has indolent growth. Surgical resection of the breast prothesis is often curative. In infiltrative, disseminated BIA-ALCL with disease involvement beyond the breast prothesis, the clinical course and prognosis appear similar to systemic ALK-negative ALCL and should be treated accordingly. Further investigations are warranted to establish effective and early treatment [29,30] (Figure 4).

### 3.4. Hepatosplenic Gamma Delta T-Cell Lymphoma: (HSTL)

Hepatosplenic gamma delta T-cell lymphoma (HTSL) is a very rare, lethal subtype of TCL-NHL, accounting for less than 1% of NHL [31]. Although the pathogenesis of HSTL is not completely understood, up to 20% of cases of are seen in long-term immunosuppressed patients, most often post-organ transplantation and in patients receiving immunosuppressive agents for autoimmune disorders such as Crohn disease and rheumatoid arthritis [32]. The origin of HSTL in most instances has been hypothesized to be related to immature cytotoxic γδ T cells residing in the spleen. The exact function of this cell is still unclear; however, it is believed to act as the hinge between both the innate and adaptive immune system. It is hypothesized that the pathophysiology of HTSL arises from upregulation of those cytotoxic γδ T cells via the JAK/STAT pathway, induced by various immunomodulator therapies such as azathioprine and monoclonal antibodies that inhibit TNF-α inhibitors (such as infliximab) [33,34]. Histologically, the malignant lymphocytes show a spectrum of features, such as cytoplasm devoid of azurophilic granules, nuclei with condensed chromatin and inconspicuous nucleoli, high nucleus-to-cytoplasm ratio, and fine chromatin. HSTL cells can often resemble lymphoblasts encountered in acute leukemia. Malignant lymphocytes initially involve and infiltrate the sinusoids and cords of the red pulp of the spleen. Uncontrolled proliferation within the red pulp causes atrophy of the white pulp.

Additional sites include the sinusoids of liver and bone marrow. At the time of presentation, 100% of all patients will have bone marrow infiltration. Specific immunophenotype features include positivity of CD3, TIA-1, Granzyme B, and TRCT γδ markers, and negativity of CD5-, CD4-/CD8-, and CD56 markers [33,35]. HTSL usually affects middle-aged patients who present with symptoms typical of lymphoma, including fever, weight loss, and night sweats, as well as fatigue, abdominal pain, and sometimes jaundice [31,35]. Imaging findings include hepatic and splenic infiltration by HSTL with sparing of the portal triads and splenic white pulp in the form of increased metabolic activity on FDG PET/CT as well as increased osseous uptake. The hallmark of the disease is lack of systemic or localized lymphadenopathy, despite extensive disseminated disease. Given the aggressive course of the disease, radiologists should always include HTSL in their differential diagnosis list, specifically in patients who clinically present with suspected underlying hematologic malignancy and proper FDG PET/CT findings. Hepatosplenomegaly with lack of significant lymphadenopathy and normal FDG uptake are unique imaging findings that differentiate HSTL from other lymphomas. Due to significant pancytopenia encountered in those patients, due to marrow infiltration or marrow toxic effects from chemotherapy, ofttimes those patients receive granulocyte-macrophage colony-stimulating factor (GM-CSF) which can mimic HSTL findings. A bone or liver biopsy is frequently necessary to establish the diagnosis and may be guided by imaging findings of disease at these sites [31] (Figure 5).

### 3.5. Peripheral T-Cell Lymphoma Not Otherwise Specified NOS: (PTCL-NOS)

Peripheral T-cell lymphoma, not otherwise specified (PTCL-NOS) represents a heterogeneous group of mature T-cell lymphomas that do not fit in any of the specific T-cell lymphoma categories. In the new 2016 WHO classification, PTCL-NOS is categorized as a unique group of aggressive T-cell neoplasms [5]. In western countries, PTCL-NOS accounts for only 4% of all NHL cases and most common types of MTLs, compared to a higher incidence in Asia, where it accounts for 20% and 25% of NHL and MTCL, respectively [36].

PTCL-NOS is diagnosed on an exclusion basis as the disease lacks any other features that fit PTCL subtypes. It is believed to originate from T-helper 2. Even with novel developed immunophenotyping and advanced laboratory techniques for molecular marker labelling, 30–50% of PTCL cases are not yet classifiable and are categorized as PTCL-NOS. Since the improvement of gene expression and molecular markers, about 15% of cases previously diagnosed as PTCL-NOS were reclassified as angioimmunoblastic lymphoma. Pathologically, PTCL-NOS are characterized by a mixture of medium and large T-cell lymphocytes. Studies showed certain abnormalities associated with PTCL-NOS such as *TET2, IDH2* and CD28 markers. PTL-NOS can be differentiated from AITL and ALCL by gene expression profiling (GEP) [5,37]. Most PTCL-NOS patients present with cutaneous symptoms in the form of patches, papules, nodules, and ulcerations. Presence of ulcerated lesions, lesion multifocality, and the presence of B symptoms usually indicates systemic involvement of PTCL-NOS [19] (Figure 6).

The course of the PTCL-NOS is usually aggressive and includes cutaneous involvement in addition to systemic dissemination in the form of extensive lymphadenopathy with liver, gastrointestinal (GI) tract, spleen, and bone marrow involvement (Figure 7). On FDG PET/CT, PTCL-NOS lesions tend to be FDG-avid, with the possible exception of cutaneous lesions and bone-marrow involvement which may be FDG-occult, knowledge of such pitfalls is crucial for both the physician and radiologist. In such instances, determining the bone-marrow status is important for the prognosis of these patients, especially in the presence of lymphopenia and elevated LDH which may necessitate biopsy and pathology correlation to exclude infiltration. In disseminated disease, PTCL-NOS is indistinguishable from other subtypes of aggressive disseminated lymphoma [19,31,36].

### 3.6. Angioimmunoblastic T-Cell Lymphoma (AITL)

Angioimmunoblastic T-cell lymphoma (AITL) is now a well-established subtype of mature peripheral T-cell lymphoma (PTCL). Many names have been previously designated to this entity before the currently recognized nomenclature identified by the 2008 World Health Organization classification of lymphoid neoplasms and the expanded 2016 revision. Previous names include angioimmunoblastic lymphadenopathy with dysproteinemia, immunoblastic lymphadenopathy, and lympho-granulomatosis X. All are predecessor synonyms to AITL and often forgotten in systemic reviews [5,38].

AITL originates from T follicular CD4 cells, and accounts for less than 1% of all NHL cases and 20% of MTCL cases. This disease was initially considered an abnormal immune response but has now been accepted as a subtype of MTL [39]. The 2016 WHO classification of lymphoid neoplasms recently acknowledged the complexity of this diagnosis with the addition of other AITL-like subsets. AITL now resides under the umbrella of nodal T-cell lymphomas with the follicular T-helper phenotype [5,40]

AITL represents almost 20% of all diagnosed cases of MTCL annually. AITL primarily affects late-to-middle-aged persons with a median age at presentation of 65 years. Contrarily to other subtypes of MTCL which are more common in Asia, ATIL shows higher incidence in European populations [5].

Clinically, most patients present with B symptoms and generalized lymphadenopathy. Nodal disease is often mild in extent with low bulk disease (1.5–3 cm) and variable uptake on FDG PET/CT. Nearly 90% of patients will present with advanced disease, and more than 70% of patients will have bone marrow infiltrative disease at the time of presentation [40,41,42]. The condition is associated with aggressive behavior and is more commonly seen in the elderly population. Imaging features are nonspecific and are similar to those of disseminated lymphoma. PET/CT usually displays high SUV, correlating to aggressive tumor behavior. AITL is associated with a high rate of relapse with a low survival rate [40] (Figure 8).

### 3.7. Enteropathy-Associated T-Cell Lymphoma of the Bowel (EATL)

Most lymphomas affecting the gastrointestinal (GI) tract are B-cell in origin, such as diffuse large B-cell lymphoma (DLBCL), mantle cell, marginal zone, and follicular lymphomas. TCLs of the GI tract are exceedingly rare. A unique subtype of the latter is known as enteropathy-associated T-cell lymphoma, which primarily involves the proximal jejunum and ileum, and represents a diagnostic challenge.

In the recent 2016 WHO classification, this entity has been revised to include three subgroups, making make each of these diseases their own entity. The two aggressive variants are now called enteropathy-associated T-cell lymphoma EATL, previously known as EATL type I, and monomorphic epitheliotropic intestinal T-cell lymphoma (MEITL), previously known as EATL type II. The third subtype, indolent T-cell lymphoproliferative disorder of the GIT has a more indolent course [5,43]. EATL is an extremely rare T-cell neoplasm, representing less than 1% of cases of NHL. EATL is often seen in patients with refractory celiac disease and is more commonly seen in European descendants. EATL is derived from intestinal intraepithelial lymphocytes (IEL) which undergo uncontrolled proliferation in response to gluten protein. On the contrary, MEITL is not associated with celiac disease and is more commonly seen in Asia. MEITL is characterized by a monomorphic proliferation of lymphoid cells with CD8 and CD56 expression which are mostly derived from γδ T cells [5,44]. EATL is approximately 5–10 times more common than MEITL. EATL often presents with substantial mixed infiltrates of eosinophils and histocytes and coagulative necrosis. The mucosa adjacent to the tumor usually shows enteropathy in the form of villous atrophy and crypt hyperplasia. Additionally, infiltrates within the lamina propria, and intraepithelial lymphocytosis can be observed. T-cell lymphocytes are often CD 4- & CD 8-negative with coexpression of CD30. Nevertheless, MEITL tumoral infiltrate is characterized by clumping of round uniform cells, showing dark nuclei and a halo of pale cytoplasm. A mix of tumoral and inflammatory cells often infiltrate the intestinal crypt epithelium. Cells often display CD56 and CD8 positivity [43]. Microscopically, MEITL often lack of coagulative necrosis, with normal surrounding mucosa [5,22].

The medial age of presentation is 60 years, with more preferential involvement of men than women. Clinically, patients present with nonspecific GI tract symptoms, such as abdominal pain, weight loss, nausea, and vomiting, but rarely bowel obstruction. More than 90% of cases develop in the small intestine. The proximal jejunum and ileum are the most common sites, with less involvement of other sites. Unlike B-cell lymphomas (BCLs) of the GI tract, which present with nonobstructing masses, TCLs of the GI tract present with diffuse or focal segmental ulcerative mucosal lesions, raised plaques or nodules [45,46]. Compared to BCLs of the bowel, prognosis of TCLs is much poorer, with 5-year survival rates of less than 20%. Extraintestinal involvement can also be rarely seen [31,47] (Figure 9).

Diagnosis is often delayed and 70% of patients often present with advanced disease due to complications. Therefore, patients with history of celiac disease presenting with refractory GI tract symptoms, despite adequate treatment, and without identified abnormality on both upper and lower endoscopy (infection, masses), should have further workup. Different modalities include video capsule endoscopy, double balloon enteroscopy, 18F FDG PET/CT as well as MRI enterography [48,49].

Differential possibilities for monoclonal T-cell infiltrative lesions of the small bowel include inflammatory bowel diseases such as Crohn’s disease, certain B-cell lymphomas, and others disseminated T-cell lymphomas. This can be further differentiated by immunohistochemical studies. BCLs of the GI tract can readily be identified by the presence of B-cell antigens such as CD20. Other disseminated T-cell lymphomas can also be identified. As an example, natural killer (NK)/T-cell lymphoma, nasal type, may show lymphocytes with CD 56 positivity. Unlike MEITL, these lymphomas are usually negative for CD8 and positive for Epstein–Barr virus. Disseminated ALCL may also involve the GI tract, however, more commonly it involves the stomach rather than the small bowel, and often shows large lymphoid anaplastic cells with characteristic CD30-positive immunostaining compared to EATL [47,50].

## 4. Conclusions

The updated fourth edition WHO classification of lymphoid malignancies released in 2016 contains important information for both radiologists and oncologists. Certain rare lymphomas may be occasionally encountered in clinical practice. Although some of those disorders have distinct clinical and imaging features, many overlap with more common disorders, thus causing delay in diagnosis and management. However, early diagnosis of many of these disorders is key as many of these are potentially treatable and early intervention may be lifesaving.

## Figures and Tables

**Figure 1 cancers-13-05217-f001:**
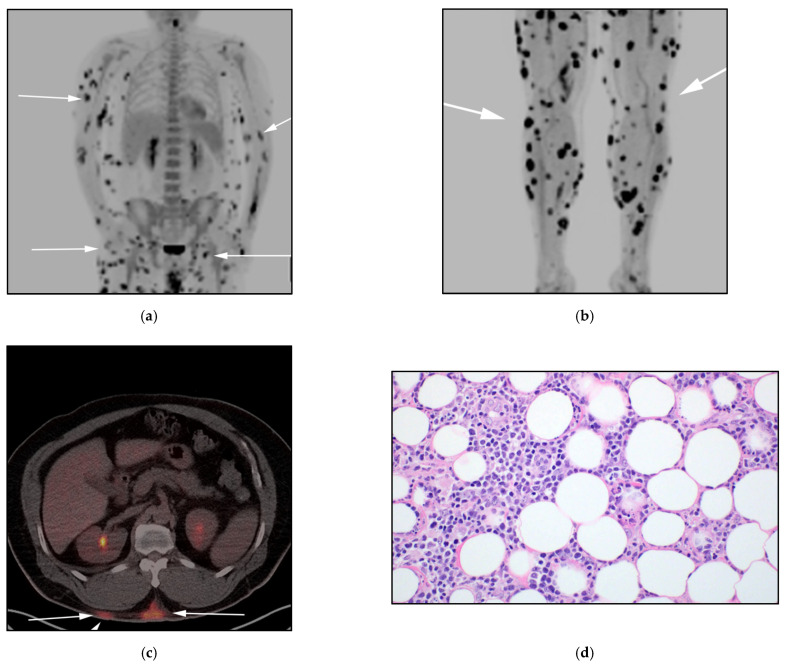
A 50-year-old-male presenting with multiple diffuse erythematous papules, biopsy revealed subcutaneous panniculitis T-cell lymphoma (STPCL). (**a**,**b**) F-18 FDG PET/CT maximal intensity projection (MIP) image, revealing extensive hypermetabolic cutaneous and subcutaneous lesions within the trunk as well as the limbs (white arrows). (**c**) F-18 FDG PET/CT axial, shows hypermetabolic nodular mass in the mid posterior abdominal wall (white arrows). (**d**) Histologic section from subcutaneous tissue shows monotonous medium-to-large neoplastic lymphoid cells encircling fat lobules in a lacelike distribution. The neoplastic cells show hyperchromatic nuclei. Note the vacuolated macrophages containing karyorrhectic debris. (Original magnification 400×, H&E stain).

**Figure 2 cancers-13-05217-f002:**
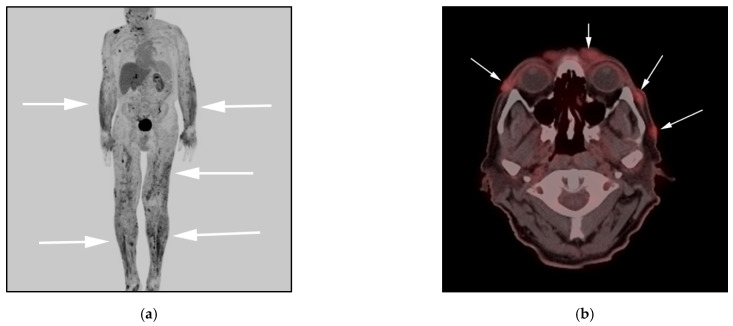
(**a**) F-18 FDG PET/CT maximal intensity projection (MIP) image in a different patient with diagnosis of primary cutaneous ALCL, ALK-negative variant, shows hypermetabolic diffuse dermal and subcutaneous nodularity extending from the top of head throughout bilateral upper arms, upper thighs and lower feet (white arrows). (**b**) F-18 FDG PET/CT axial images show hypermetabolic subcutaneous nodules/masses in the head, and anterior face (white arrows).

**Figure 3 cancers-13-05217-f003:**
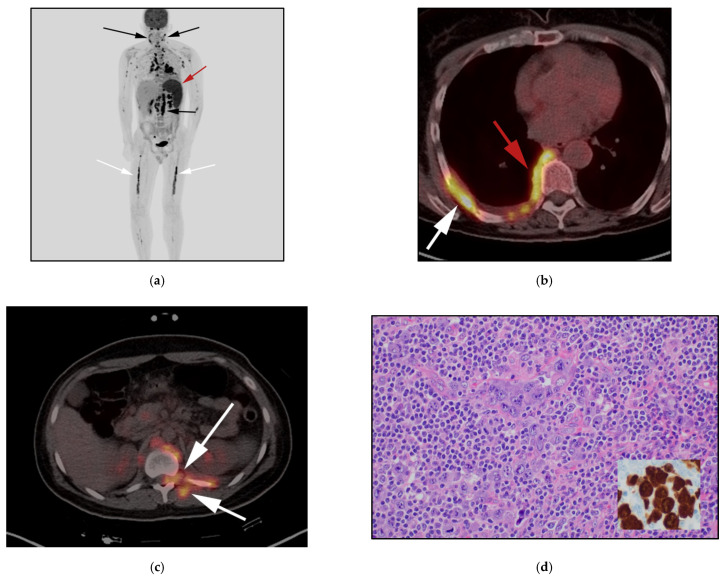
A 22-year-old male diagnosed with systemic ALCL, ALK-negative variant. (**a**) F-18 FDG PET/CT maximal intensity projection (MIP) image, showing diffuse hypermetabolic adenopathy throughout the neck, chest, and abdomen (black arrows), diffuse osseous activity within lower extremities, consistent with marrow infiltration (white arrows). Additionally, note increased activity within the spleen, consistent with splenic infiltration (red arrows). (**b**) F-18 FDG PET/CT axial image in a different patient, shows hypermetabolic circumferential right pleural mass (red arrows) associated with right seventh rib-destructing mass (white arrow). (**c**) 18 FDG PET/CT axial image in a different patient, shows extensive hypermetabolic soft-tissue masses along left 12th rib with extension into the T12-L1 left neural foramen and epidural space, consistent with perineural spread. This again was biopsy-proven systemic ALCL, ALK-negative variant (white arrow). (**d**) Histologic section shows cohesive sheets of tumor cells with anaplastic features resembling nonhematopoietic metastatic tumors of the lymph node. Note the large hallmark cells showing eccentric kidney shaped nuclei. The tumor cells strongly express CD30 and ALK-1 expression. (Original magnification 400×, H&E stain).

**Figure 4 cancers-13-05217-f004:**
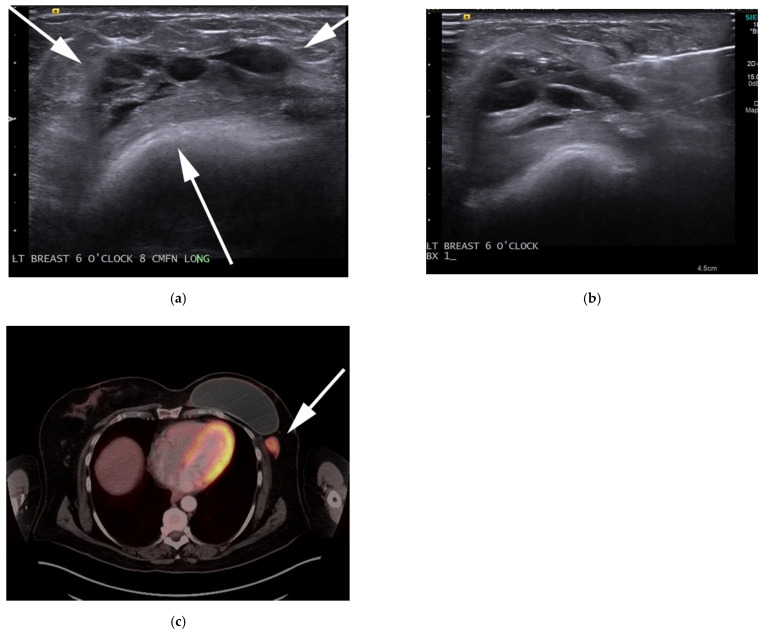
A 57 -year-old woman with left breast enlargement 8 years after textured implant placement. (**a**) Ultrasound of the left breast implant shows preimplant complex solid and cystic mass (two superior arrows), the inferior arrow shows echogenicity of the breast implant. (**b**) Ultrasound of the left breast shows core needle biopsy of this complex solid and cystic mass. (**c**) Axial FDG PET/CT in another 72-year-old woman who had a palpable left axillary node, F-18 FDG PET/CT axial image shows hypermetabolic left axillary node (white arrows) related to left breast implant. No preimplant masses or fluid collections seen. Both cases were biopsy-proven to be breast-implant anaplastic large T-cell lymphoma. Although ultrasound is the most sensitive modality for detecting peri-implant effusion, FDGPET/CT is essential for detecting hypermetabolic capsular masses as well as systemic disease spread.

**Figure 5 cancers-13-05217-f005:**
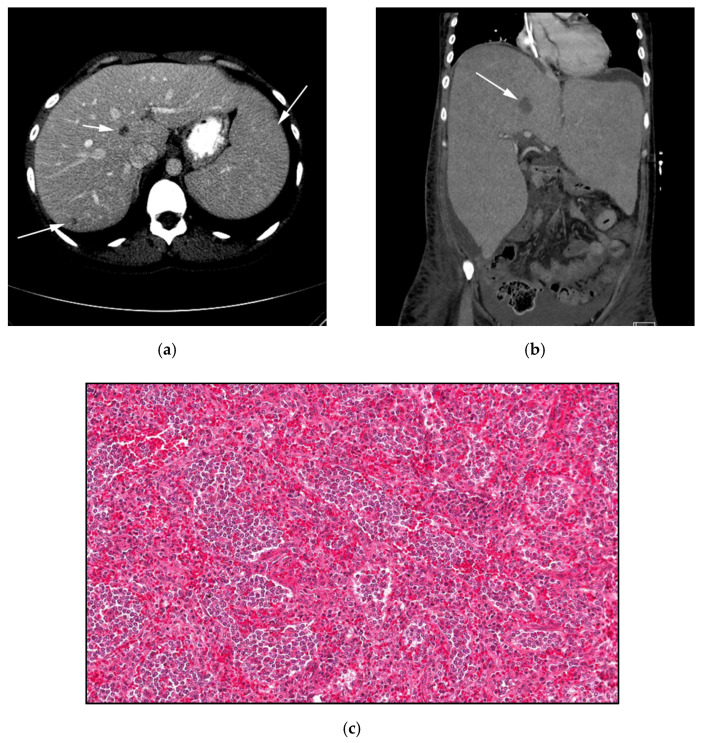
A 31-year-old woman with known history of hepatosplenic gamma delta T-cell lymphoma (*γδHST*). (**a**) Axial CECT shows splenomegaly and two small indeterminate focal hepatic lesions (white arrows). Patient had biopsy-proven γδHSTL after splenectomy. (**b**) Coronal CECT obtained three years later shows marked increased size of liver reaching up to 28 cm in craniocaudal dimensions. There is also increased size of the previously seen central focal hepatic lesion (white arrow). (**c**) Histologic section shows moderate-sized neoplastic cells with condensed nuclei, small nucleoli and a pale cytoplasmic rim. The neoplastic cells are infiltrating the red pulp of the spleen but show a striking predilection to the splenic sinusoids. In contrast, *γδHST* spares the portal triads and the splenic white pulp. (Original magnification 400×, H&E stain).

**Figure 6 cancers-13-05217-f006:**
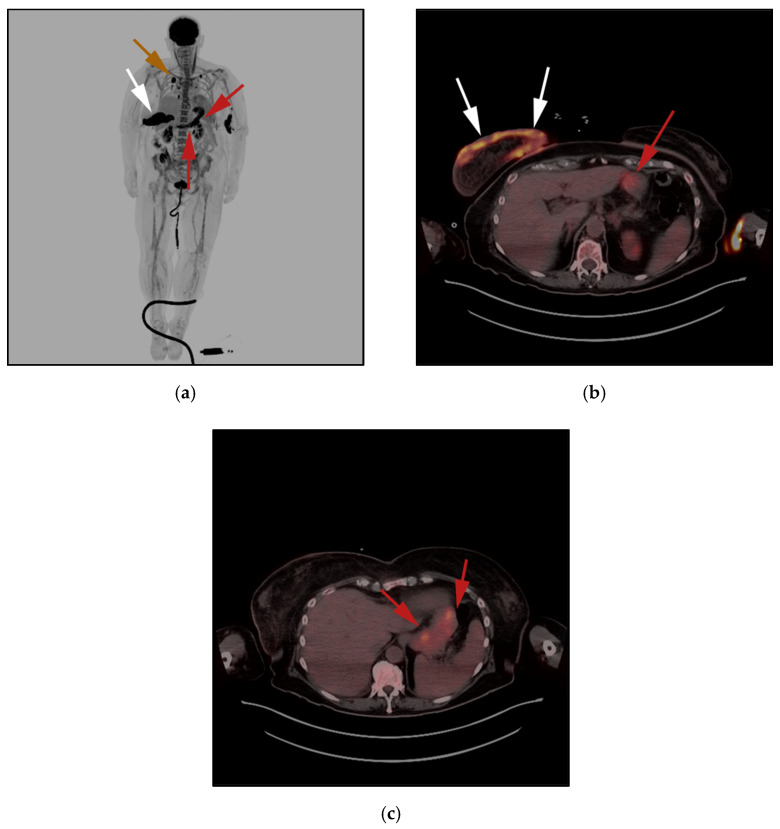
50-year-old-female with known diagnosis of PTCL-NOS. (**a**) F-18 FDG PET/CT maximal intensity projection (MIP) image demonstrates increased activity localized to the anterior chest wall (white arrow), curvilinear activity seen within the epigastric region (red arrows) as well as rounded activity within the right upper lung (brown arrow). (**b**,**c**) F-18 FDG PET/CT axial images confirm the location of the lesion within the skin of the right breast (white arrow) as well as stomach (red arrows).

**Figure 7 cancers-13-05217-f007:**
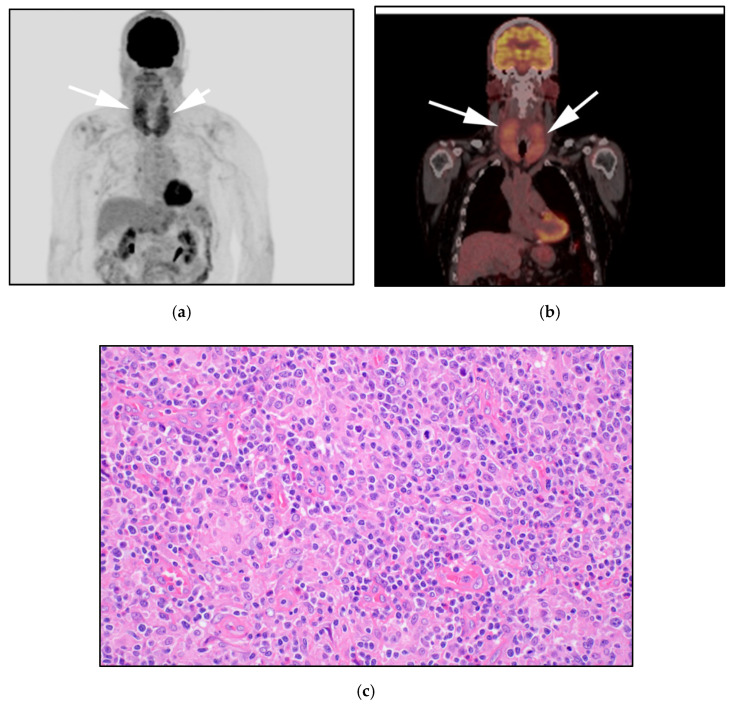
71-year-old-man with known diagnosis of PTCL-NOS. (**a**) F-18 FDG PET/CT maximal intensity projection (MIP) image demonstrates diffuse hypermetabolic butterfly activity within the mid neck (white arrows). (**b**) FDG PET/CT axial images confirm activity localized to diffusely hypermetabolic enlarged thyroid glands (white arrows). Note no suspicious marrow lesions seen. Marrow biopsy of this patient showed T-cell infiltration, despite negative marrow involvement on PET/CT (which has high rate of false negativity for marrow infiltration). (**c**) Histologic section shows, that cells are atypical medium-to-large-sized cell infiltrates that efface the lymph node normal architecture. A variable number of eosinophils are noted. There is also a prominent epithelioid histiocytic component. (Original magnification 400×, H&E stain).

**Figure 8 cancers-13-05217-f008:**
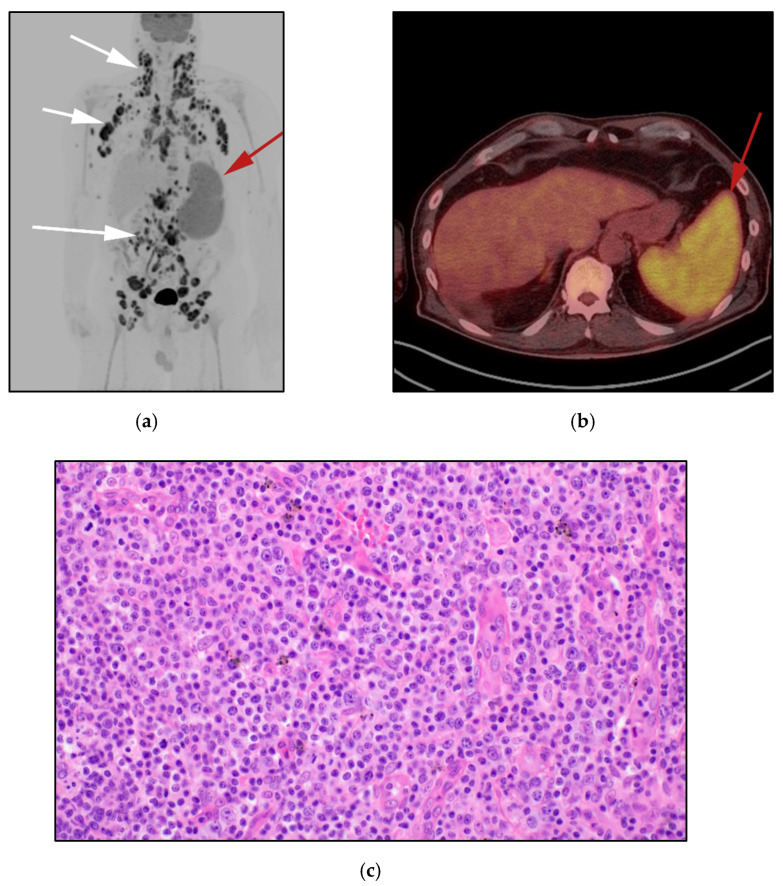
A 77-year-old-male with known diagnosis of AITL (**a**) F-18 FDG PET/CT maximal intensity projection (MIP) image reveals extent of generalized adenopathy throughout the neck, chest, abdomen and pelvis (white arrows) and hypermetabolic splenomegaly (red arrow), consistent with splenic infiltration. (**b**) Axial images shows increased activity and size of the spleen. (**c**) Histologic section shows, atypical diffuse infiltrate of pleomorphic lymphocytes composed of small- and medium-sized cells with rare eosinophils. The neoplastic cells have pale-to-clear cytoplasm. Arborizing blood vessels are evident. (Original magnification 400×, H&E stain).

**Figure 9 cancers-13-05217-f009:**
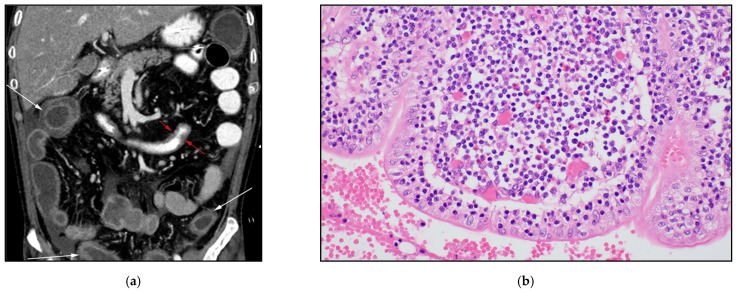
60-year-old-male with known diagnosis of Crohn’s disease, presenting with refractory symptoms in the form of abdominal pain, watery diarrhea and weight loss (**a**) Coronal CECT images show extensive thickening of the large and small bowel in the form of submucosal edema and mucosal enhancement (white arrows). Additionally, note mucosal polypoidal thickening of the proximal jejunal bowel loops (red arrows). Patient underwent endoscopy and biopsy of jejunal bowel loops, revealing primary T-cell lymphoma of the bowel. (**b**) The histologic section discloses type II EATL from a patient presenting with an intestinal lymphoma with villous blunting and goblet cell depletion, with atypical intraepithelial lymphocytes showing prominent epitheliotropism. Note the monotonous population of medium-sized lymphoid cells with slightly dispersed chromatin, inconspicuous nucleoli and ample clear cytoplasm, imparting a monocytoid appearance in the submucosa. (Original magnification 400×, H&E stain).

**Table 1 cancers-13-05217-t001:** 2016 WHO classification of mature T-cell and natural killer cell (NK) neoplasms, with highlights of molecular, immunophenotype and clinical features of most common subtypes.

Histologic Subtype	Molecular Change & Characteristic Immunophenotype	Clinical Characteristic
T-cell prolymphocytic leukemia T-cell large granular lymphocytic leukemia Chronic lymphoproliferative disorder of NK cells Aggressive NK-cell leukemia Systemic EBV1 T-cell lymphoma of childhood Hydroa vacciniforme-like lymphoproliferative disorder		
Adult T-cell leukemia/lymphoma	HTLV-1 infection leads to the clonal expansion and immortalization of CD4+ T cells and CD 24+ cells Transformation of Tax and HBZ, modulating signal transduction pathways (CREB/ATF, NF-κB, JAK-STAT, mTOR-AKT	More endemic in Japan 2/3 cases present as leukemia, 1/3 as lymphoma Organomegaly and skin involved.
Extranodal NK-/T-cell lymphoma, nasal type	Clonal episomal EBV infection of NK or T cells Overexpression of cytotoxic proteins (granzyme H) and PDGFRA.	Occurs in nasopharynx (nasal type) and less frequently at other anatomic sites (extranasal)
Enteropathy-associated T-cell lymphoma Monomorphic epitheliotropic intestinal T-cell lymphoma Indolent T-cell lymphoproliferative disorder of the GI tract	HLA DQ2 or DQ8 haplotypes. TCRγδ phenotype, and a lack of TCR expression Not fitting EATL or MEITL	Associated with celiac Not associated with celiac Ulcerative bowel mass
Hepatosplenic T-cell lymphoma	It is most commonly of γδ _T-cell lineage Isochromosome 7q [i(7)(q10)] is reported to be a frequent recurrent chromosomal aberration	History of immune dysfunction Hepatosplenomegaly with lack of adenopathy
Subcutaneous panniculitis-like T-cell lymphoma	T-cell infiltration of the subcutaneous fat without dermal or epidermal involvement	Prolonged course of recurrent panniculitis Hemophagocytic syndrome
Mycosis fungoides (MF) Sézary syndrome (SS)	Most common cutaneous subtypes Memory T-lymphocyte (CD45RO+)	MF: Localized skin SS: Skin and dissmentated disease.
Primary cutaneous CD30 1 T-cell lymphoproliferative disorders Lymphomatoid papulosis Primary cutaneous anaplastic large-cell lymphoma Primary cutaneous gd T-cell lymphoma		
Primary cutaneous CD8+ aggressive epidermotropic cytotoxic T-cell lymphomas		
Primary cutaneous acral CD8+ T-cell lymphoma	New provisional entity	Described in ear
Primary cutnaeous CD4 1 small/medium T-cell lymphoproliferative disorder		
Peripheral T-cell lymphoma, NOS	Most common subtype Lacks specific morphologic and phenotypic features of other PTCL subtypes Associatedwith translocation t(5:9)(q33:32)	Most commonly cutaneous symptoms B symptoms in systemic involvement
Follicular T-cell lymphoma		
Angioimmunoblastic T-cell lymphoma	2nd most common subtype Associated with immune dysregulation of T-follicular helper (TFH) cells Recurrent gains of chromosomes 3q, 5q, and 21	Generalized lymphadenopathy B symptoms
Nodal peripheral T-cell lymphoma with TFH phenoytpe		
Anaplastic large-cell lymphoma, ALK+ and ALK-subtypes	ALK gene located on chromosome 2p23, with the nucleophosmin gene (NPM), on 5q35 Overexpression of MYC i	Can be differentiated based on initial site of presentation: systemic ALCL, primary cutaneous ALCL, and breast implant-associated ALCL (BIA-ALCL). ALK+ has better clinical course compared to ALK-variant
Breast implant–associated anaplastic large-cell lymphoma	New entitiy A subtype of ALK + ve cases Cells express CD30 and frequent marker for cytotoxic T-cells.	Excellent prognosis Two forms, either preimplant effusion or axillary adenopathy

Sources—[11].

**Table 2 cancers-13-05217-t002:** Summary of changes contained in the updated 2016 WHO classification of T-cell lymphoid malignancies. New entities have been included, former entities have been reclassified, and some provisional entities have been “promoted” to reflect the advancement of our understanding of lymphoma molecular biology and clinical behavior.

Entiry/Category	Updates	New Diagnostic Considerations
Subcutaneous panniculitis-like T-cell lymphoma (SPTCL)	No updates	No updates
Hepatosplenic delta T-cell lymphoma (HSTCL)	No updates	No updates
Angioimmunoblastic T-cell lymphoma (AITL)	New subtypes	Follicular T-cell lymphoma subtype
Enteropathy-associated T-cell lymphoma (EATL)	Previously designated EATL type 1	Diagnosis only to be used for cases formerly known as type I EATL, typically linked to celiac disease (CD) More common in northern European descent
Monomorphic epitheliotropic intestinal T-cell Lymphoma (MEITL)	Formerly known as EATL type 2 segregated from type I EATL and given a new name due to its distinctive nature	Formerly type II EATL Not linked to celiac disease More seen in Asian population
Indolent T-cell lymphoproliferative disorder of the GI tract	Subtype of Primary T-cell lymphoma of the bowel	A recently introduced indolent provisory disorder with monoclonal T-cell infiltrate of the superficial layer of the bowel, progression was noted in some cases
Primary cutaneous acral CD8+ T-cell lymphoma	New provisional entity	Originally described as originating in the ear.
Peripheral T-cell lymphoma not otherwise specified (PTCL-NOS)	New subgroups identified	Novel subsets classified on basis of molecular and genetic anomalies, are not currently part of routine clinical service, but maybe ultimately clinically recognized
ALK anaplastic large-cell lymphoma (ALCL)	ALK, +ve, no updates ALK, −ve, new subgroups	Subset of ALK −ve, now a definite entity that includes cytogenetic subsets that appear to have prognostic implications (e.g., 6p25 rearrangements at IRF4/DUSP22 locus).
Breast implant-associated anaplastic large-cell lymphoma (BIA-ALCL)	New provisional entity	Distinguished from other ALK2 ALCL −ve; noninvasive disease associated with excellent outcome.

Sources—[4,5].

**Table 3 cancers-13-05217-t003:** Suggested clinical classification of mature T-cell lymphomas based on initial site of presentation.

Extranodal	Nodal	Skin	Disseminated
Hepatosplenic T-cell lymphoma (HSTCL)	Peripheral T-cell lymphoma, not otherwise specified (PTCL-NOS)	Mycosis fungoides (MF) Sezary Syndrome (SS)	Adult T-cell Leukemia/Lymphoma (ATLL)
Extranodal T-cell lymphoma—nasal type (EKNTL)	Systemic anaplastic large-cell lymphoma (ALCL-ALK variants)	Subcutaneous Panniculitis-like TCL (SPTCL)	T-prolymphocytic Leukemia (T-PLL)
Enteropathy-associated T-cell lymphoma (EATL)		Primary cutaneous ALCL (PCALCL)	T-cell large granular lymphocytes (LGL)
Monomorphic, epitheliotropic, intestinal T-cell lymphoma (MEITL)		Primary cutaneous CD4+ small/medium pleomorphic T-cell lymphoma (PCSM-TCL)	Aggressive natural killer T-cell leukemia
Breast implant-associated ALCL (BIA-ALCL)			

Sources—[4,7].

**Table 4 cancers-13-05217-t004:** Key clinical, imaging, and pathologic features of some mature T-cell lymphomas (MTCLs).

Entity	Key Pathologic Features	Key Clinical Features	Key Imaging Features	Differential Considerations
Subcutaneous panniculitis-like T-cell lymphoma (***SPTCL***)	T-cell infiltration of the subcutaneous fat without dermal or epidermal involvement	Red or violaceous subcutaneous nodules Erythematous plaques. Ulcerated skin lesions involving trunk, extremities	Enhancing, hypermetabolic subcutaneous nodules/masses	*Sezary syndrome (SS)* and mycosis fungoides (MF) Primary cutaneous gamma/delta T-cell lymphoma Inflammatory skin nodules such as systemic lupus Cutaneous metastasis from breast carcinoma, melanoma
Anaplastic large-cell lymphoma (**ALCL**)	Subdivided based on ALK-kinase enzyme ALK+ disease has better prognosis	Cutaneous subtype presents with cutaneous plaque nodules Systemic subtype presents with disseminated disease and virtually any organ can be involved	Generalized lymphadenopathy Bone marrow infiltration Soft tissue masses in lungs, mediastinum, pleura, and pericardium Gastrointestinal masses Peritoneal, omental, and mesenteric masses	Cutaneous subtype: subcutaneous panniculitis-like T-cell lymphoma Systemic diffuse metastases from an unknown primary malignancy, other aggressive lymphomas
Breast implant-associated anaplastic large-cell lymphoma (***BIA-ALCL***)	ALK-ve ALCL variant CD30 positivity of peri-implant fluid	Localized disease with excellent outcome, often seen as delayed preimplant effusion presenting with unilateral breast enlargement (most common) Palpable breast mass (less common) Axillary nodal enlargement	Delayed peri-implant fluid collection Enhancing peri-implant mass/nodule Enlarged axillary lymph nodes Uncommonly disseminated disease	Peri-implant post-surgical seroma Peri-implant hematoma Implant rupture Breast carcinoma
Hepatosplenic T-cell lymphoma (***HSTL***)	Stains positive for T-cell receptor γδ chain and leucocyte common antigen Stains negative CD20 (positive in diffuse large B-cell lymphoma)	Immunocompromised patients. B symptoms., i.e., night sweats, fatigue, fever Organomegaly including liver/spleen	Hepatosplenomegaly Lack of lymphadenopathy	Other subtypes of lymphoma, i.e., primary hepatic lymphoma or disseminated diffuse large B-cell lymphoma
Peripheral T-cell lymphoma, not otherwise specified (**PTCL-NOS**).	Most common MTCL subtype Majority peripheral T-cell lymphoma that could not be classified based on immunophenotypic and molecular marker will be diagnosed as not otherwise specified subtype Considered diagnosis of exclusion Category for unclassifiable peripheral T-cell lymphomas	No specific clinical findings	Extensive lymphadenopathy Reticuloendothelial system involvement, liver, spleen, and bone marrow involvement Preferential cutaneous, breast and GI tract involvement.	Other aggressive lymphomas
Angioimmunoblastic T-cell lymphoma (**AITL**)	Second most common subtype of MTCL Originates from T-helper follicular cell	Older age Extensive nodal involvement, organomegaly Cutaneous involvement in the form of rash, pruritis	PET/CT sensitivity for cutaneous involvement is almost 100%	Other aggressive lymphomas
Enteropathy-associated T-cell lymphoma of the bowel (**EATL**)	Enteropathy-associated T-cell lymphoma (EATL) previously known as EATL type I Monomorphic epitheliotropic intestinal T-cell lymphoma (MEITL), previously known as EATL type II. Indolent T-cell lymphoproliferative disorder of the GIT	EATL associated with celiac disease (CD) MEITL no relation to celiac disease	PET/CT more sensitive than CT for detecting diffuse or focal segmental bowel lesions in the form of extensive mucosal ulceration	Celiac disease Inflammatory bowel disease

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
