# Peer review of "Uncommon Variants of Mature T-Cell Lymphomas (MTCLs): Imaging and Histopathologic and Clinical Features with Updates from the Fourth Edition of the World Health Organization (WHO) Classification of Lymphoid Neoplasms"

_cancers, 2021, doi:10.3390/cancers13205217_

Round 1
Reviewer 1 Report
The authors described rare variant T-cell lymphomas in the updated 4th WHO classification of lymphoid malignancies, showing PET-CT image and histology. Summary of the images may help diagnosis of the rare variants. You should revise the following points.
- Primary cutaneous acral CD8 T-cell lymphoma was mistyped in table1.
- A line of ALCL-ALK variants was duplicated in table2.
- The full term of PCSM-TCL should be described as Primary Cutaneous CD4+ Small/Medium Sized Pleomorphic T Cell Lymphoma in table 2.
Author Response
Reviewer 1:
The authors described rare variant T-cell lymphomas in the updated 4th WHO classification of lymphoid malignancies, showing PET-CT image and histology. Summary of the images may help diagnosis of the rare variants. You should revise the following points.
- Primary cutaneous acral CD8 T-cell lymphoma was mistyped in table1.
Thank you for your feedback, this has been changed.
- A line of ALCL-ALK variants was duplicated in table2.
Thanks for the feedback, the duplicated row has been removed.
- The full term of PCSM-TCL should be described as Primary Cutaneous CD4+ Small/Medium Sized Pleomorphic T Cell Lymphoma in table 2.
Thank you for the feedback, this has been changed.
Reviewer 2 Report
I think that this is a very comprehensive review in the field of hemolymphopathology.
No comments or anything to add.
Author Response
Thank you so much for your feedback.
Reviewer 3 Report
This manuscript is a review that focuses on the clinicopathological characteristics and imaging of Mature T- and NK- cell neoplasms, in particular to uncommon variants of mature T-cell lymphomas (MTCLs). The review is well written, it is easy to read, and have enough tables and references. First, an introduction is made, and the tables show the characteristics of the different lymphoma subtypes. Later, the authors focuses on the description of some of the uncommon variants.
Major comments (but it is up to the authors to accept this comments for improvement or not):
1) The authors could make a table with all the Mature T- and NK- cell neoplasms, exactly as appear in the current WHO 2016 classification.
2) A table with the most clinical, immunophenotype, and molecular change that are characteristic of each subtype could also be included.
3) Please be aware that a new 5th revised classification will appear relatively soon. I attach the draft in this reply.
Minor comments:
1) Lines 65-66: “Various T-cells include T-helper (Th) cells, natural killer (NK) cells, suppressor T cells, cytotoxic, memory cells, γδ T‐cells [4].” Could you please confirm that NK cells are within the T cells subtype? [HSC àCLP, CLP cells develop to (1) Bcell, (2) T cell lineages, (3) helper ilc lineages, and (4) NKP, iNK, and finally mNK…]
2) Table 1. “Primary cutaneous acral CD81 T-cell lymphoma”. Could you please correct the name of the entity? The correct name is “Primary cutaneous acral CD8+ T―cell lymphoma”
3) Table 1. “ALK2 anaplastic large-cell lymphoma (ALCL)”. Could you please use “ALK” (number 2 is not necessary).
4) Line 118. “with the term mature T-cell lymphomas (MTCLs)”. If I am not wrong, I think that the exact term is “Mature T- and NK-cell neoplasms”.
5) Lines 121. “The purpose of this article is to descry”. Could you please correct to spelling to “describe”? You could use a free tool for checking the text known a “Trinka AI (please google it)”
6) Line 212. “distinguished from other ALK2 ALCL”. Comment: “2” ???
7) Lines 341-342. Could you please write the names of the genes in italics?
8) Line 386. “classi-fication”. Could you please delete the “-“. You could use the abbreviation WHO in all the manuscript as it is quite known word.
9) Lines 427-428. Please correct the word “EATL-LEATL” to “EATL”.
10) Line 435. “with possible CD20 expression”. Could you please confirm that it is correct to say that the cells could express a B-cell marker such as the CD20?
11) Figure 9. The authors are showing a case of EATL that affected a patient with Crohn’s disease. Would it be better to show a case of a patient with Celiac disease (the characteristic association)?

Author Response
Reviewer 3: This manuscript is a review that focuses on the clinicopathological characteristics and imaging of Mature T- and NK- cell neoplasms, in particular to uncommon variants of mature T-cell lymphomas (MTCLs). The review is well written, it is easy to read, and have enough tables and references. First, an introduction is made, and the tables show the characteristics of the different lymphoma subtypes. Later, the authors focuses on the description of some of the uncommon variants.
Major comments (but it is up to the authors to accept this comments for improvement or not):
1) The authors could make a table with all the Mature T- and NK- cell neoplasms, exactly as appear in the current WHO 2016 classification.
2) A table with the most clinical, immunophenotype, and molecular change that are characteristic of each subtype could also be included.
Comment number (1) and (2): The authors would like to thank reviewer #3 about this suggestion.
A new table (Table 1) summarizing mature T-cell and NK neoplasms has been created. This table lists all mature T- and NK-cell neoplasms, exactly as it appears in 2016 classification. Additionally, this table highlights of molecular, immunophenotype and clinical features of most common subtypes.
Table 1. 2016 WHO classification of Mature T-cell and Natural killer cell (NK) neoplasms, with highlights of molecular, immunophenotype and clinical features of most common subtypes.
|
Histologic subtype
|
Molecular change & Characteristic immunophenotype |
Clinical characteristic
|
|
· T-cell prolymphocytic leukemia · T-cell large granular lymphocytic leukemia · Chronic lymphoproliferative disorder of NK cells Aggressive NK-cell leukemia · Systemic EBV1 T-cell lymphoma of childhood · Hydroa vacciniforme-like lymphoproliferative Disorder
· Adult T-cell leukemia/lymphoma
· Extranodal NK-/T-cell lymphoma, nasal type
· Enteropathy-associated T-cell lymphoma Monomorphic epitheliotropic intestinal T-cell lymphoma · Indolent T-cell lymphoproliferative disorder of the GI tract
· Hepatosplenic T-cell lymphoma
· Subcutaneous panniculitis-like T-cell lymphoma
· Mycosis fungoides (MF) · Se ́zary syndrome (SS)
· Primary cutaneous CD30 1 T-cell lymphoproliferative disorders · Lymphomatoid papulosis · Primary cutaneous anaplastic large cell lymphoma · Primary cutaneous gd T-cell lymphoma · Primary cutaneous CD8 1 aggressive epidermotropic cytotoxic T-cell lymphomas
· Primary cutaneous acral CD81 T-celllymphoma
· Primary cutnaeous CD4 1 small/medium T-cell lymphoproliferative disorder
· Peripheral T-cell lymphoma, NOS
· Follicular T-cell lymphoma
· Angioimmunoblastic T-cell lymphoma
· Nodal peripheral T-cell lymphoma with TFH phenoytpe
· Anaplastic large-cell lymphoma, ALK+ and ALK-subtypes
· Breast implant–associated anaplastic large-cell lymphoma |
· HTLV-1 infection leads to the clonal expansion and immortalization of CD4+ T cells and CD 24+ cells · Transformation of Tax and HBZ, mopdulating signal transduction pathways (CREB/ATF, NF-κB, JAK-STAT, mTOR-AKT
· Clonal episomal EBV infection of NK or T cells · Overexpression of cytotoxic proteins (granzyme H) and PDGFRA.
· HLA DQ2 or DQ8 haplotypes. · TCRγδ phenotype, and a lack of TCR expression · Not fitting EATL or MEITL
· It is most commonly of γδ _T-cell lineage · Isochromosome 7q [i(7)(q10)] is reported to be a frequent recurrent chromosomal aberration
· T-cell infiltration of the subcutaneous fat without dermal or epidermal involvement
· Most common cutaneous subtypes · Memory T-lymphocyte (CD45RO+)
· New provisional entity
· Most common subtype · lacks specific morphologic and phenotypic features of other PTCL subtypes · Associatedwith translocation t(5:9)(q33:32)
· 2nd most common subtype · Associated with immune dysregulation of T-follicular helper (TFH) cells · Recurrent gains of chromosomes 3q, 5q, and 21
· ALK gene located on chromosome 2p23, with the nucleophosmin gene (NPM), on 5q35 · Overexpression of MYC i
· New entitiy · A subtype of ALK+ve cases · Cells express CD30 and frequent marker for cytotoxic T-cells. |
· More endemic in Japan · 2/3 cases present as leukemia, 1/3 as lymphoma · Organomegaly and skin involvement.
· Occurs in nasopharynx (nasal type) and less frequently at other anatomic sites (extranasal)
· Associated with celiac · Not associated with celiac · Ulcerative bowel mass
· History of immune dysfunction · Hepatosplenomegaly with lack of adenopathy
· prolonged course of recurrent panniculitis · Hemophagocytic syndrome
· MF: Localized skin · SS: Skin and dissmentated disease.
· Described in ear
· Most commonly cutaneous symptoms · B-symptoms in systemic involvement
· Generalized lymphadenopathy · B-symptoms
· Can be differentiated based on initial site of presentation: systemic ALCL, primary cutaneous ALCL, and breast implant associated ALCL (BIA-ALCL). · ALK+ has better clinical course compared to ALK- variant
· Excellent prognosis · Two forms, either preimplant effusion or axillary adenopathy
|
3) Please be aware that a new 5th revised classification will appear relatively soon. I attach the draft in this reply.
Thanks for the feedback, I was wondering how can this new classification be cited if it’s not published yet?
Minor comments:
- Lines 65-66: “Various T-cells include T-helper (Th) cells, natural killer (NK) cells, suppressor T cells, cytotoxic, memory cells, γδ T‐cells [4].” Could you please confirm that NK cells are within the T cells subtype? [HSC àCLP, CLP cells develop to (1) Bcell, (2) T cell lineages, (3) helper ilc lineages, and (4) NKP, iNK, and finally mNK…]
Thanks for the feedback, this has been added and highlited within the text.
- Table 1. “Primary cutaneous acral CD81 T-cell lymphoma”. Could you please correct the name of the entity? The correct name is “Primary cutaneous acral CD8+ T―cell lymphoma”
Thanks for the feedback, this has been corrected.
3) Table 1. “ALK2 anaplastic large-cell lymphoma (ALCL)”. Could you please use “ALK” (number 2 is not necessary).
Thanks for the feedback, this has been changed.
4) Line 118. “with the term mature T-cell lymphomas (MTCLs)”. If I am not wrong, I think that the exact term is “Mature T- and NK-cell neoplasms”.
Thanks for the feedback, this has been changed.
5) Lines 121. “The purpose of this article is to descry”. Could you please correct to spelling to “describe”? You could use a free tool for checking the text known a “Trinka AI (please google it)”
Thanks for the feedback, this has been changed.
6) Line 212. “distinguished from other ALK2 ALCL”. Comment: “2” ???
Thanks for the feedback, the number 2 has been removed.
7) Lines 341-342. Could you please write the names of the genes in italics?
Thanks for the feedback, this has been changed to italics format.
8) Line 386. “classi-fication”. Could you please delete the “-“. You could use the abbreviation WHO in all the manuscript as it is quite known word.
Thanks for the feedback, this has been changed.
9) Lines 427-428. Please correct the word “EATL-LEATL” to “EATL”.
Thanks for the feedback, this has been changed,
10) Line 435. “with possible CD20 expression”. Could you please confirm that it is correct to say that the cells could express a B-cell marker such as the CD20?
Thanks for the feedback, this sentence has been removed from the text.
11) Figure 9. The authors are showing a case of EATL that affected a patient with Crohn’s disease. Would it be better to show a case of a patient with Celiac disease (the characteristic association)?
Unfortunately, I couldn’t find a case with EATL secondary to Celiac disease.
